



# A social-ecological approach to identify and quantify biodiversity tipping points in South America's seasonal dry ecosystems

Kirsten Thonicke[1,2], Fanny Langerwisch[1,2], Matthias Baumann[3], Pedro J. Leitão[3,4], Tomáš Václavík[5,6], Ane Alencar[7], Margareth Simões[8,9], Simon Scheiter[10], Liam Langan[10], Mercedes Bustamante[11], Ignacio Gasparri[12], Marina Hirota[13,14], Jan Börner[15], Raoni Rajao[16], Britaldo Soares-Filho[16], Alberto Yanosky[17], José-Manuel Ochoa Quinteiro[18], Lucas Seghezzo[19], Georgina Conti[20], Anne Cristina de la Vega-Leinert[21]

[1]Earth System Analysis, Potsdam Institute for Climate Impact Research (PIK), P.O. Box 60 12 03, D-1412 Potsdam, Germany
[2]Czech University of Life Sciences Prague (CULS), Kamýcká 129, 165 00 Praha 6 – Suchdol, Czech Republic
[3]Geography Department, Humboldt-Universität zu Berlin, Unter den Linden 6, D-10099 Berlin, Germany
[4]Department Landscape Ecology and Environmental System Analysis, Technische Universität Braunschweig, Langer Kamp 19c, D-38106 Braunschweig, Germany
[5]Department of Computational Landscape Ecology, UFZ-Helmholtz Centre for Environmental Research, Permoserstraße 15, D-04318 Leipzig, Germany
[6]Department of Ecology and Environmental Sciences, Faculty of Science, Palacký University Olomouc, Šlechtitelů 27, 78371 Olomouc, Czech Republic
[7]O Instituto de Pesquisa Ambiental da Amazônia (IPAM), Bairro Asa Norte, Brasilia-DF, 70863-520, Brazil
[8]Empresa Brasileira de Pesquisa Agropecuária (EMBRAPA), Soil Institute, Rio de Janeiro, Brazil
[9]Rio de Janeiro State University UERJ/FEN/DESC/PPGMA, Rio de Janeiro, Brazil
[10]Senckenberg Biodiversity and Climate Research Centre (BiK-F), Senckenberganlage 25, D-60325 Frankfurt am Main, Germany
[11]Instituto de Ciências Biologicas, Universidade de Brasília, Campus Universitário Darcy Ribeiro - Asa Norte, 70910 Brasília, Brazil
[12]Instituto de Ecología Regional, Conicet - Universidad Nacional de Tucumán, Argentina
[13]Department of Physics, Federal University of Santa Catarina, Florianópolis, Brazil
[14]Institute of Biology, University of Campinas, Campinas, Brazil
[15]Institute for Food and Resource Economics and Center for Development Research, University of Bonn, Bonn, Germany
[16]Centro de Sensoriamento Remoto, Universidade Federal de Minas Gerais, Av. Antônio Carlos, 6627, Belo Horizonte, Brazil
[17]Asociación Guyra Paraguay - CONACYT Paraguay, Asunción, Paraguay
[18]Instituto de Investigación de Recursos Biológicos Alexander von Humboldt, Bogotá, Colombia
[19]Instituto de Investigaciones en Energía No Convencional, CONICET, Universidad Nacional de Salta, Salta, Argentina
[20]Instituto Multidisciplinario de Biología Vegetal, Consejo Nacional de Investigaciones Científicas y Técnicas, Universidad Nacional de Córdoba, Facultad de Ciencias Exactas Físicas y Naturales, IMBiV (CONICET-UNC), Córdoba, Argentina
[21]Institute of Geography and Geology, University Greifswald, Greifswald, Germany

*Correspondence to*: Kirsten Thonicke (Kirsten.Thonicke@pik-potsdam.de)

**Abstract.** Tropical dry forests and savannas harbour unique biodiversity and provide critical ecosystem services (ES), yet they are under severe pressure globally. We need to improve our understanding of how and when this pressure provokes tipping points in biodiversity and the associated social-ecological systems. We propose an approach to investigate how drivers leading to natural vegetation decline trigger biodiversity tipping and illustrate it using the example of the Dry Diagonal in South America, an understudied deforestation frontier.

The Dry Diagonal represents the largest continuous area of dry forests and savannas in South America, extending over three million km² across Argentina, Bolivia, Brazil, and Paraguay. Natural vegetation in the Dry Diagonal has been undergoing large-scale transformations for the past 30 years due to massive agricultural expansion and intensification. Many signs indicate that natural vegetation decline has reached critical levels. Major research gaps prevail, however, in our understanding of how these transformations affect the unique and rich biodiversity of the Dry Diagonal, and how this affects the ecological integrity and the provisioning of ES that are critical both for local livelihoods and commercial agriculture.



Inspired by social-ecological systems theory, our approach helps to explain:

(i) how drivers of natural vegetation decline affect the functioning of ecosystems, and thus ecological integrity,

(ii) under which conditions, where, and at which scales the loss of ecological integrity may lead to biodiversity tipping

points, and

(iii) how these biodiversity tipping points may impact human well-being.

Implementing such an approach with the greater aim of furthering more sustainable land use in the Dry Diagonal requires a transdisciplinary collaborative network, which in a first step integrates extensive observational data from the field and remote sensing with advanced ecosystem and biodiversity models. Secondly, it integrates knowledge obtained from dialogue

processes with local and regional actors as well as meta-models describing the actor network. The co-designed methodological framework can be applied not only to define, detect, and map biodiversity tipping points across spatial and temporal scales, but also to evaluate the effects of tipping points on ES and livelihoods. This framework could be used to inform policy making, enrich planning processes at various levels of governance, and potentially contribute to prevent biodiversity tipping points in the Dry Diagonal and beyond.

## 65  1 Introduction

Multiple drivers of natural vegetation decline related to agricultural expansion affect the functioning of the Dry Diagonal, especially the Cerrado and Dry Chaco biomes in South America, and put their biodiversity at risk. Continued habitat losses increase the risk of biodiversity tipping points (BD-TPs) which would affect social-ecological systems (SES, Ostrom, 2009), incl. the well-being of local (traditional) and regional social systems. Ecological integrity (EI, Andreasen et al., 2001), which

describes the status of structure (fragmentation), ecosystem function and species composition, can be used to quantify the risk related to BD-TP in multi-functional ecosystems and allows for the detection of cascading tipping points.

More specifically, the social and environmental drivers which result in the degradation of EI and thereby increase the risk for BD-TP need to be understood in order to investigate how such sudden shifts affect the provision of ecosystem services to people in the Dry Diagonal. Capturing such an impact chain requires the development of a methodological framework

following an inter- and transdisciplinary approach. Natural and social scientists need to work side by side and share their complementary expertise on climate change, land cover and land-use change, species diversity, landscape fragmentation and ecosystem function, ecosystem services, and related, multi-facetted societal and cultural drivers and impacts of social-ecological transformations at local and regional level. To avoid biodiversity tipping points or distal linkages in ES supply, these different expert groups have to join forces to explore possible policy incentives and nature conservation instruments.

To overcome the challenges involved in promoting a constructive communication and transfer information (data, meta-data, contextual information) across disciplines and socio-cultural contexts within science and at the science-policy interface, activities that foster team and trust-building, the co-design of the research and communication process and the creation of discussion platforms play a critical role (e.g. joint scenario development, storytelling, collective writing, cf. Moser, 2016).

In this paper we summarize the outcome of such an interdisciplinary exercise in developing a methodological framework to

identify and detect BD-TP for the deforestation frontiers of the Dry Diagonal. We review existing literature to describe the current knowledge basis for each framework component. The following section describes the current status of natural vegetation decline in the Dry Diagonal, and state-of-the-art knowledge on detecting BD-TPs. Section 2 then introduces the methodological framework to detect BD-TPs based on changes in EI components and how this could help to assess BD-TP impacts on human well-being. Section 3 provides a short outlook on potential applications and its use to inform policy and

planning processes to reduce the risk of tipping.



## 1.1 Natural vegetation loss in the Dry Diagonal

Tropical dry forests and savannas cover roughly 20% of the global land surface, contribute to 30% of the global primary productivity, sustain about 20% of the human population, and harbour astonishing levels of biodiversity, including many endemic species (c.f. Baldi et al., 2015; Baumann et al., 2017a; Lehmann, 2010; Miles et al., 2006; Murphy et al., 2004).

These ecosystems also sustain the livelihood of millions of people, including many indigenous or traditional communities but also industrial-scale agriculture producers, while providing regionally and globally important ecosystem services (ES), e.g. food production and water provision (IPBES, 2018). Dry forests and savannas also experience very high anthropogenic pressure, especially through land-use change and concomitant ecosystem transformation, but also from overexploitation and climate change (Jobbágy et al., 2015; Miles et al., 2006; Parr et al., 2014). Despite their importance for human well-being

and their outstanding conservation value, recently research started to focus more on these systems (Banda-R et al., 2016; Kuemmerle et al., 2017; Parr et al., 2014)(Cerri et al., 2018; Spera et al., 2016). This lack of knowledge undermines efforts to balance human resource use with EI preservation, the conservation of ES, and the protection of local livelihoods.

The largest continuous area of dry forests and savannas is the Dry Diagonal in South America, spanning across three million square kilometres from Argentina, through Bolivia and Paraguay, into Brazil (Figure 1). The region has recently emerged as

a global hotspot of natural vegetation decline, with rates being 2.5 higher than the one currently in the Amazon (Strassburg et al., 2017) with the expansion of intensified cropping and cattle ranching leading to massive transformation of the natural vegetation, particularly in the Dry Chaco (Northern Argentina, Paraguay and Bolivia) and the Cerrado (Brazil) (Baldi et al., 2015; Espirito-Santo et al., 2016; Klink and Machado, 2005; Parr et al., 2014; Volante et al., 2016). Although natural vegetation decline (NV decline, hereafter) in many areas in the Dry Diagonal exceeds levels that have been found to mark

critical thresholds in other ecosystems (Pardini; Swift and Hannon, 2010), its effects on biodiversity, ecological integrity, and their social-ecological outcomes remain weakly understood.

We selected the Dry Chaco and Cerrado portion of the Dry Diagonal (Figure 1) as study region (Dry Diagonal, hereafter), because it shares strong similarities in the general composition and structure of natural vegetation. Indeed, it consists of a complex, heterogeneous mosaic of ecosystems with varying levels of woody vegetation, from closed woodlands to open

grasslands via savannas. Although the Dry Diagonal has distinct evolutionary history, both in comparison to other forest ecosystems in South America, and tropical forests globally (Slik et al., 2018), the Dry Chaco and Cerrado ecoregions in are exceptionally biodiverse (species richness and endemism) and are at risk of unprecedented biodiversity loss under ongoing land-use and potentially also future climate change. Alone the Cerrado counts about 10,000 vascular plant species, over 800 species of birds, about 200 species of mammals, and over 100 species of amphibians (Mendonça et al., 2008; Ratter et al.,

1997; Veiga et al., 2005).

In the study region, the main actor groups shaping the landscape and leading to the large-scale transformations of NV decline are large-scale agribusinesses and cattle ranchers who have expanded in the region since the late 1980s, but especially since the year 2000. The main land-use forms are intensified pasture and silvopastoral (savanna-like) systems, and arable cultivation (in particular corn, soybean and cotton) (Diniz-Filho et al., 2009; Zak et al., 2008). At the same time, the region

includes a wide variety of small-scale farmers, who practice mixed farming, and small homesteads that practice subsistence farming and woodland grazing (Baldi et al., 2015; Cáceres et al., 2015; Eloy et al., 2016; Piquer-Rodríguez et al., 2018). Fuelwood extraction and charcoal production additionally alter the structure and composition of natural vegetation (Cáceres et al., 2015; Ratter et al., 1997).

The vast majority of NV decline is due to soybean expansion and the conversion into extensive grazing areas, and, to a lesser

extent, charcoal production. Land-use change is mainly driven by large-scale agribusinesses in some parts of the region, whereas subsistence smallholders and indigenous communities dominate northern parts. A wide variety of indigenous and traditional communities, live in the Dry Diagonal including, e.g. the quilombolas, the descendants of former fugitive slaves settlements in the Cerrado. Their livelihoods depend on subsistence farming, collection and hunting of non-timber forest



products, fuelwood collection and water provision which requires ecologically intact ecosystems. However, these actors are

increasingly replaced by large-scale agribusiness creating a trade-off between local and global provision of ES, and endangering nature conservation efforts by these indigenous communities (Baldi et al., 2015; Eloy et al., 2016).

The Dry Diagonal is exposed to large-scale transformation in land-use, the rates and temporal dynamics of which differ substantially regionally. The Chaco has lost 30% of its natural vegetation over the past 30 years to agricultural practices (Vallejos et al., 2015), however multiple factors may explain recent slowing down in deforestation rates (Volante and

Seghezzo, 2018). The Cerrado lost 34% of its natural vegetation until mid-1990s (Ratter et al., 1997), and cumulatively amounting to 46% of natural vegetation cover loss  t an alarming rate of 1% p.a. to date (Strassburg et al., 2017). Land-use transitions differ markedly between countries, e.g. expansion of intensified ranching in Paraguay, but soybean agriculture in Argentina and Bolivia (Sano et al., 2010).

The Dry Diagonal extends over different countries, which are also characterized by diverse conservation and land-use

policies. For example, Argentina disposes of  land-use zoning, which restricts land-use practices in most of the Chaco region, whereas Brazilian and Paraguayan legislation has no appreciable land use restrictions, which has resulted in up to 80% of NV decline in the Chaco and Cerrado (Seghezzo et al., 2011; Soares-Filho et al., 2014). While 70 protected areas have been designated in the Bolivian lowlands and Yungas regions by 2013 (cumulating to 23,2 million of hectares), i.e. 30% of the total surface of lowlands and Yungas in Bolivia are protected (Naturaleza, 2016), and less than 1.2% of the Argentinian Dry

Chaco (Fehlenberg et al., 2017) and 7.5% of the Brazilian Cerrado are protected public lands (Strassburg et al., 2017). The continued establishment of protected areas, such as the 105,000 ha Traslasierra National Park in Argentina (https://www.parquesnacionales.gob.ar/2018/03/nuevo-parque-nacional-traslasierra/) are important developments for conserving the Dry Chaco as a habitat. However, more often than not, these protected areas are effectively "paper parks", where land use restrictions cannot be enforced due to lack of appropriate resources (Watson et al., 2014; de la Vega-Leinert

and Huber, 2019). To capture the specific mechanisms resulting in natural vegetation decline in each of the Dry Diagonal countries, in-depth analysis of the particular socio-economic contexts and legislative frameworks that drive or fail to regulate deforestation and land use change are required.

Decadal and interannual climate variability, increasingly also climate change, influences vegetation dynamics. Changes in the continent's atmospheric circulation can create a precipitation dipole where parts of south-eastern South America are

affected by drought and the other by intensive rainfall (Vera and Díaz, 2015). Additionally, long-term decadal changes have led to an increase in historic precipitation in the southern Cerrado and in the Chaco. The region moreover receives precipitation from the Amazon region through continental moisture transport, which currently contributes about 27% of the annual precipitation in the Dry Diagonal (Zemp et al., 2014). Continued tropical deforestation could lead to self-amplified vegetation loss in the Amazon (Zemp et al., 2017) and also further affect continental moisture transport and thus climate in

the Dry Diagonal. For South-eastern South America, mean annual temperatures are projected by nearly 2°K and mean annual precipitation possibly increase between 1 to 7% in South-eastern South America (Christensen et al., 2013). It is still not fully understood how this will affect the structure and composition of natural vegetation and how climate change will contribute to increase the risk of BD-TPs.

Initial evidence points towards the existence of critical tipping points in biodiversity within the Dry Diagonal. For example,

avian biodiversity in the Dry Chaco changes non-linearly along gradients of land-use intensity, exhibiting remarkable resilience up to critical levels of land-use intensity beyond which biodiversity loss accelerates drastically (Macchi et al., 2019). Similar nonlinear biodiversity change has been identified for bats in the Cerrado (Roque et al., 2018), mammals in the Gran Chaco (Periago et al., 2014), specifically jaguar (Quiroga et al., 2013), birds in the Dry Chaco (Macchi et al., 2019) and damselfly communities (Rodrigues et al., 2016). More broadly, vertebrates endemic to the Gran Chaco show performance

curves that strongly decline the lower the size of protected areas are (Nori et al., 2016). Those non-linear changes could accelerate NV decline or changes in vegetation composition via changes in seed dispersal (Periago et al., 2014).



Understanding where and when BD-TP in relation to natural vegetation loss have been – or will be – crossed will improve our knowledge basis for policy making and planning, while providing deep scientific insights into tipping points across scales.

**1.2 Biodiversity tipping points in multi-functional social-ecological systems**

Ecosystems are multi-functional and embedded in social-ecological systems (SES, Ostrom, 2009). Ecosystems are coupled to social systems via the supply of ecosystem services, while social systems (e.g. institutions, businesses, communities, households) influence ecosystems and drive NV decline, or nature protection, through for example strategic management, daily practice, worldviews and cultural values, intake of knowledge, technological change (Erb, 2012; Hummel, 2008; Liehr

et al., 2017). Through these coupling flows ecosystems and social systems co-evolve. In this study, we describe ecosystems using the EI concept bearing in mind that climate influences the functioning of co-evolving social-ecological systems.

BD-TPs can occur due to single or combination of external and internal driving factors. It makes it therefore necessary to use a flexible conceptual framework that captures the impact of NV decline on any possible drastic and/or rapid loss in biodiversity. The EI concept enables the assessment of the structural, compositional and functional changes at the ecosystem

and the species level (Andreasen et al., 2001; Wurtzebach and Schultz, 2016). We propose to use these terms as follows: *Structure* refers to the spatial and vertical organization of ecosystems and landscapes, as well as the distribution patterns of species. *Composition* refers to the diversity of ecosystems and of species (alpha, beta and gamma diversity). *Function* refers to key functional aspects of ecosystems and species, such as primary productivity, carbon storage, or predation (e.g., Midgley, 2012). The EI concept therefore allows us to adequately analyse the complex impact of NV decline on biodiversity

while avoiding pitfalls of focusing solely on a single metric. By systematically exploring how respective EI metrics react to NV decline, the temporal and spatial scales of potential BD-TP can be identified. Because EI metrics can be linked to ES, the impact of BD-TP on (local and regional) livelihoods can be described. However, the societal implications of BD-TP and ES loss are often place and situation dependent. Therefore, to capture the consequences of BD-TP for the SES under study and derive generic insights of relevance for policy and management, detailed, contextualized, case study approaches, e.g. based

on policy analysis and local ethnographies, play a critical role.

The challenge is to define tipping points for the ecosystem component of the SES while acknowledging the flows from society to the ecosystem via for example policy, management, societal preferences, knowledge and practices (Liehr et al., 2017). So far, tipping points were defined for ecosystems or the Earth System (Lenton, 2013) to quantify non-linear abrupt changes. Two viewpoints exists in describing the systems behaviour to an impact: 1) engineering resilience that allows the

ecosystem to return to the initial condition, and 2) ecological resilience, which focusses on the capacity of an ecosystem to withstand an impact (Bahn and Ingrisch, 2018). However, to capture the dynamics of a wide indicator list, a broad definition of tipping points is required. We use van Nes et al. (37, p. 904) definition of tipping points as a *"situation where accelerating change caused by a positive feedback drives the system to a new state"*. We focus on non-linear changes in the ecosystem, where a rapid and sudden NV decline leads to a drastic EI change, which can be mapped along the driving

variable of change (NV decline, Figure 2a) or by exploring single EI metrics over time (Figure 2b). This allows for identification of resistance against, absorption of, and recovery from an impact (green, yellow and light-green areas in Figure 2b).

Ecosystems can react in different ways to changes in the conditions, such as climate and NV decline (i.e. drivers). We propose to follow the concept of ecological resilience which incorporates the absorption of, and the recovery from, an impact

(Figure 2b). *Resistant* ecosystems maintain their function despite the occurrence of an impact (Grimm and Wissel, 1997; Gunderson, 2000), i.e. there is no change measurable in an ecosystem state variable, here described by a EI metric V. In contrast, we regard an ecosystem to be *resilient* when it manages to recover its pre-impact EI $V_{pre}$ with a range of average recovery rates $V_{ave}$ (Figure 2b) and within the average recovery time. Where or when full recovery is not possible, a net



change might lead to a modified ecosystem state (green arrows in Figure 2b), allowing the system to have multiple stable
states, e.g. $V_2$ and $V_3$ (cf. Gunderson, 2000; Nimmo et al., 2015). The latter criterion is important because it acknowledges
the variability of the ecosystem embedded in long-term transient processes such as climatic changes, succession and
evolution. An ecosystem has tipped when it cannot recover a substantial portion of its pre-impact state $V_{pre}$ within half of the
average recovery time (dashed red line and orange box in Figure 2b, (cf. Mitchell et al., 2016)). Here, the tipping point
would be the critical state $V_{crit}$ for the EI metric V. While a critical level of NV decline, $NV_{crit}$, can lead to a sudden and rapid
loss in an EI metric to the critical state $V_{crit}$ (Figure 2a), secondary impacts which act as disturbance events and system-
internal dynamics can lead to feedbacks and change $V_{crit}$ and further reduce the critical threshold for NV decline to $NV_{crit,E}$
(Dakos et al., 2012; Lenton et al., 2008). Climate change modifies disturbance impacts which cause water deficits, changed
fire regimes and grazing pattern. These disturbance impacts form second-order effects for which further attributes can be
defined that are decisive for the ecosystem's ability to resist, absorb or recover from the impact. It depends on the precise
condition of the studied ecosystem to define $V_{crit}$ and $V_{crit,E}$ for a given NV decline and climate change condition, e.g. in the
Dry Diagonal, and postulate it as the BD-TP.

Whether our approach can also be used to understand the resilience of society, more precisely local farmers and indigenous
communities is still open to debate. Resilience is increasingly been considered an important element of more sustainable SES
(Berkes et al., 2003). The theory of resilience was first described for natural ecosystems (Holling, 1973) but was later
extended to human systems (e.g. cities, communities, and individuals Davidson et al., 2016), and to the relationships
between humans and the environment (Folke, 2006; Holling, 2001) to encompass social-ecological resilience (Folke et al.,
2016). Arguably, to be useful in decision-making processes, societal resilience should be amenable to some kind of
quantitative or qualitative translation (Carpenter et al., 2014). However, detecting and describing impacts of tipping points
and the resulting resilience in a coupled SES is a challenge because of the many possible interactions between the numerous
interacting components they comprise. Thus, for example changes in global and regional demand for resources drive NV
decline, i.e. land-use change or overexploitation, which can lead to non-linear feedbacks in the ecosystem and affect
ecosystem service provision to society. To disentangle these feedback loops, a linear approach that starts from the ecosystem
perspective and describes the impacts of BD-TP for the ecosystem, i.e. through EI, and how the flow from the ecosystem to
society is affected, i.e. provision of ES, is required.


ES can be used to the changes in the coupling flow between ecosystem and society and describe the implication for society if
nature's provision to society is severely disturbed. Therefore, non-linear relationships and trade-offs between EI, ES and
resulting human well-being at the local and regional scale can be a starting point. Crossing BD-TPs can even lead to the
destabilization of societies, which can be the case in the dry forests of the Dry Diagonal, where agricultural expansion
promotes social conflicts among landholders with different ES preferences, as well as among the local and indigenous
communities (Cáceres et al., 2015; Seghezzo et al., 2011). Here, culture plays as important role in defining nature-society
interaction and emphasize and operationalize the role of indigenous and local knowledge in understanding nature's
contribution to people (Diaz et al., 2018) which can possibly be captured in individual case studies.

The identification, contextualization and quantification of local ES demand is context-dependent and requires
complementary case-studies complemented by a systematic review of published studies of relevant social-ecological
transformations in the Dry Diagonal. This can include local settings, where traditional livelihood strategies are threatened by
the expansion of commercial soy and cattle production, or where alternative livelihood strategies associated with emerging
sectors (e.g. ecotourism) and commodities (e.g. non-timber forest product extraction) and payment for ES programmes, in
and around protected areas. In this respect, rapid rural appraisal complemented by stratified farm-household and village
surveys can contribute to understand, map and quantify local communities' reliance on ES provided by the environment.
Stratification criteria can include factors commonly associated with varying degrees of environmental dependence, such as



market integration, distance to natural vegetation frontiers, and cultural background (Angelsen, 2014). Moreover, participatory approaches can be useful to assess the sustainability of local agricultural practices and gauge the effect of changes in the provision of ES on the resilience of local and regional production systems (Mónica Liliana Vega, 2015).

Knowledge on how different actor (groups) influence NV decline and ES demand are indeed important to understand the social-ecological dimension of BD-TP. Rapid and profound loss of ES supply will affect actor groups differently, thereby exacerbating inequalities of access to natural resources and entrenching existing conflicts. Here, social networks analysis and multi-criteria evaluation approaches, such as those established for water-footprint analysis (Arjen Y. Hoekstra, 2011) could be applied to identify all relevant actors and their relative power. More insight on local actors' perspectives, preferences and

underlying value systems can further help to better understand which adaptation strategies may be socially desirable, acceptable and politically enforceable (Huaranca, 2019).

## 2 Methodological framework to identify biodiversity tipping points in multi-functional SES

To identify biodiversity tipping points and to understand their implications for nature and society a methodological

framework is required that quantifies the drivers causing NV decline, the changes in EI with possible tipping point behaviour and impacts on ES provision. The development of such a methodological framework requires expertise from natural and social scientists of different disciplines, e.g. ecologists, biodiversity experts, remote-sensing experts, rural sociologists, cultural geographers, anthropologists, ecological economists and political scientists. To co-develop the framework for the social-ecological context of the Dry Diagonal 3 regional workshops were conducted where scientists from Brazil, Argentina,

Colombia, Paraguay and Germany shared their research experience. Here, the scientific challenge to describe the ecological as well as sociological implications of BD-TP in a balanced manner became evident in terms of how to deal with research and data gaps for the Dry Diagonal, but also how to design a comprehensive interdisciplinary methodology to investigate the consequences of BD-TP for ES provision and local and regional livelihoods. We describe the methodological framework (Fig. 3) in this section.

### 285 2.1 Impacts of natural vegetation decline on ecological integrity

*Natural vegetation decline*

Climate oscillations (Vera and Díaz, 2015) and deforestation for agricultural expansion have shaped the land cover and vegetation dynamics affecting biodiversity in the Dry Diagonal (e.g., Macchi et al., 2019). The rates of NV decline have differed among countries and over time. While NV decline in the Dry Chaco has generally being increasing since the 1970s

(Vallejos et al., 2015; Volante and Paruelo, 2015), rates have recently declined in the Argentinian Dry Chaco (Volante and Seghezzo, 2018). However, natural vegetation is still lost at an alarming rate in the Paraguayan Dry Chaco (Baumann et al., 2017; Caldas et al., 2015) as well as in the Cerrado (Klink and Machado, 2005; Strassburg et al., 2017). The Dry Chaco is the largest ecoregion in Paraguay and is subject to high levels of deforestation. With more than 12,000 km², the ecoregion is being cleared for livestock production at a rate of 500-1,800 hectares/day (Marchi, 2018). Rates of deforestation and NV

decline are regularly monitored for (e.g., Arévalos, 2015; Yanosky, 2013a; Yanosky, 2013b) which negatively impacts biodiversity of the Paraguayan Chaco (Mereles, 2015). Capturing these different spatio-temporal dynamics of NV decline are an important contribution to advancing our understanding of the drivers behind NV decline.

Climate oscillations have increased annual precipitation in the Dry Chaco, resulting in increasing woody cover locally (L.E. Hoyos, 2013). While small-scale agriculture has already been present in the Chaco region for decades, the introduction of

genetically modified soy 20 years ago, triggered a massive land cover change through agricultural expansion at industrial scale (Fehlenberg et al., 2017; Grau et al., 2005; Volante et al., 2016; Volante and Paruelo, 2015). To quantify the drivers of



NV decline for the Dry Chaco, a baseline for pre-market based agriculture is required so that the transition to the industrial-scale agriculture that induced large-scale NV decline can be captured (see Fig.3, first column). This can be done using remote sensing techniques on high-resolution data (e.g., Baumann et al., 2017b) or using other geo-databases (e.g., Vallejos et al., 2015). However, to understand the spatio-temporal dynamics of those drivers, social, economic and legislative conditions need to be analysed. Although often country-specific, these factors do not operate in a vacuum or independently, which leads to spill-over or replacement effects.  For example, soybean production replaces cattle ranching which results in the acceleration of deforestation rates in other parts of the Dry Chaco (Baumann et al., 2016; Fehlenberg et al., 2017). The relationships between global demand of agricultural and forest products, international forest protection goals, national legislation and changes in NV decline at local level dynamics are intricate (see, e.g., Mills Busa, 2013) and need to be investigated in detail for particular case of the Dry Chaco.

A similar challenge exists for the Cerrado. Here, historical land-cover change needs to be mapped and pastures need to be distinguished from natural open woodlands: a process that questions the methodology of high-resolution remote-sensing techniques as currently done in the MapBiomas project (Mapbiomas, 2019). Similarly to the Dry Chaco, expansion of soybean production since the 1990ies has initiated NV decline despite the establishment of environmental policies at the same time (Eloy et al., 2016). Recent efforts towards forest protection policies nevertheless still allow for the potential legal deforestation of further 40 million ha of Cerrado (Strassburg et al., 2017). This substantially increases the risk for profound changes in local and traditional communities, including changes in traditional practices such as fire management (Eloy et al., 2016). Furthermore, fragmenting natural vegetation affects the functioning of the Cerrado ecosystem, where potential substantial impacts on its rich biodiversity are to be expected (Bustamante et al., 2012; Diniz et al., 2017). To identify the nature and direction of these changes and to quantifying them implies closing data gaps on biodiversity, vegetation dynamics and in disturbance regimes, as well as understanding how local and traditional communities may contribute to, and be affected, by these.

*Ecological Integrity*

For example, NV decline fragments habitats, which can be described by structural EI metrics (Fig. 3, second column). Fragmentation in the Cerrado does not lead to edge effects due to changes in microclimate as observed for tropical wet forest (Haddad, 2015), but opens space for invasive species such as African grasses which itself changes compositional and functional EI (Mendonca et al., 2015). Invasive grasses accelerate fire due to increased fuel flammability and increase impacts on structural and functional EI. Edge effects of fragmented natural vegetation further include changes plant litter biomass in the Cerrado (Dodonov et al., 2017). Habitat quality and the land cover characteristics surrounding the forest fragments are important for maintaining species richness. Conserving forest fragments alone will likely not halt species loss in the Chaco (Aguilar et al., 2018), even though dominant tree species could be maintained (Alves et al., 2018). Ecosystem functionality takes 15 years to recover after land use is abandoned in the Dry Chaco (Basualdo et al., 2019). Landscape fragmentation also affects compositional EI for which bird species richness and community composition could be one of the EI metric (Marini, 2001). It could be therefore expected that non-linear changes in one, e.g. structural, EI metric, is also seen in another, e.g. compositional or functional, EI metric.

Measures of EI must be based on indicators that are useful for conveying information about the composition, structure, and function of selected ecosystems over time and across spatial scales (Wurtzebach and Schultz, 2016). A more suitable alternative is to report the individual indicators of the different components of EI, usually recurring to scorecards using a traffic-light symbology relating to their status in reference to baseline conditions (Tierney et al., 2009). Respective indicators of the three EI components would have to be quantified over time for the Dry Diagonal for indicators at the ecosystem level, and at local sites for indicators at the species level. Covering the temporal dimension or the systematic exploration, along gradients of NV decline (Fig. 2a) would allow establishing relationships between NV decline and single or combined EI





metrics, and thus potential BD-TPs. Indicators to quantify changes in functional components of EI include indicators of
primary productivity, carbon stored in biomass, litter and soils, evapotranspiration which all describe vegetation dynamics
(Table 1). Using simulated plant trait distributions and their spatio-temporal changes describe the changes in functional
diversity consistent with vegetation dynamics. Data on spatial connectivity can be derived when combining habitat
information from NV decline and simulated vegetation dynamics and combining it with the indicator on functional
connectivity for key taxa in the region. While NV decline can affect vegetation dynamics and functional diversity at the
ecosystem scale, i.e. entire Dry Diagonal, changes in species composition can be mapped at large spatial scales, but require
detailed species-specific information (Table 1).

Defaunation as a consequence of climate change (Warren et al., 2018) and future land-use change (Powers and Jetz, 2019)
has been documented for thousands of species of different organism groups at the global scale. Such projections applying
Species Distribution Models, selected for functionally important or dominant taxa of the Dry Diagonal, could be used to
quantify the impact of NV decline. Cross-comparison with existing studies on mammals and birds of the Amazon
deforestation frontier (Ochoa-Quintero et al., 2015), or bird species occurrences of the Dry Chaco which declined with
decreasing woody-cover loss (Macchi et al., 2019) would be the starting point for such an exercise. EI metrics at the species
level would include data on occurrence of key taxa, their population dynamics, community turnover and richness of key taxa
(Table 1). However, species information remains anecdotal, requires new data compilation, or even more data collection in
the countries of the Dry Diagonal which would have to be improved to quantify compositional changes in EI at the species
level.

Carbon sequestration and biomass storage as well as evapotranspiration and water stored in soils are relevant ecosystem
functions which are affected by climate and land-use change. Applying flexible-trait DGVMs (Langan et al., 2017;
Sakschewski et al., 2015) to quantify respective impacts on vegetation dynamics and plant functional traits allows
quantification of functional changes in EI related to the water and the carbon cycle. Landscape fragmentation for agricultural
fields changes fire regimes and grazing which can feed back to fire via reduced fuel production (Bustamante et al., 2012).
Combining respective disturbance modules, e.g. process-based fire models optimized for the Cerrado (Drüke et al., 2019) or
herbivory effects from grazer population dynamics (Pachzelt et al., 2015; Pfeiffer et al., 2019), in flexible-trait DGVMs
allows quantification of the impact of changes in the disturbance regimes and their interaction on functional EI metrics
which could have secondary effects on BD-TPs (Table 1).

**2.2 Quantification of biodiversity tipping points based on changes in ecological integrity**

Tipping points, resilience and resistance are technical terms that aim to describe temporal changes of key elements in a
system that lead to profound changes in the functioning of the affected system. In ecology there is a long history of defining
these terms, e.g. Grimm and Wissel (1997), finding empirical evidence for resistance to disturbance, resilience, and critical
thresholds systematizing sudden changes and identify respective indicators (Dakos et al., 2014). However, biodiversity
tipping points (BD-TPs), i.e. the sudden or profound loss in biodiversity, have not been adequately defined. Building on the
van Nes definition (2016), BD-TPs would describe non-linear loss in biodiversity due to changes in driving conditions and
including positive feedbacks that would accelerate such loss ahead of the changes in the effect variable without such
feedbacks. One could also argue that the recovery of biodiversity after an impact is limited and remains below a critical
threshold (Fig. 2b). Because single or several ecosystem or biodiversity components can contribute or cause tipping, we
suggest using EI components at the species and ecosystem level. Rapid and profound changes (deep impact) of one or
several EI metrics can thus constitute a BD-TP if the recovery of one or several EI metrics remains below the critical
threshold that no longer would allow the ecosystem or species to be ecologically integer. Such EI metric changes can be
aligned to NV decline (Fig. 2a) or changes over time (Fig.2b), where the impact is the combined effect of future climate and
NV decline. To identify such break-points existing methodologies established in remote sensing can be adapted (Kennedy et





al., 2010; Verbesselt et al., 2016). With decomposing EI metric (time) series into trend, seasonal and remainder (e.g., Roque et al., 2018), slow evolving processes and abrupt events can be identified with segmenting respective data series. Early warning signals of biodiversity collapse across gradients have been quantified for tropical forest loss (Roque et al. 2018).

Early warning signals of tipping, such as a critical slowing down of the recovery rates in EI metrics, could then be based on indicators suggested by Dakos et al. (2012). However, since a decline in EI can consist of one or several indicators, multivariate statistical analysis is additionally required to identify simultaneous, delayed tipping or cascading effects (Fig. 3, 3$^{rd}$ column). We expect that such a broad-scale or top-down approach in search for biodiversity tipping points is required in diverse, highly connected (high modularity) ecosystems such as in the Dry Diagonal.


Ecosystems are multifunctional and show different levels of taxonomic, functional and structural diversity. Sudden and/or profound ecosystem changes due to NV decline will also affect the interactions between biodiversity and ecosystem function. Natural disturbances such as drought, fire, grazing or wind damage additionally influence vegetation dynamics and habitat characteristics. Changing these disturbance regimes, including their interactions, due to combined effects of climate

change and NV decline, is likely to produce secondary effects on BD-TPs. Climate change will add another level of complexity to how disturbances change the biodiversity-ecosystem function relationships. In order to explain why specific relations between NV decline and EI metric(s) might cause BD-TPs, second-order effects initiated by changes in disturbances could accelerate the occurrence of BD-TPs. Specific processes or attributes need to be included in the analysis that would quantify the resistance, impact and recovery of from these disturbances (Table 2) and need to be linked to the

tipped EI metric(s). These include attributes and processes describing the ecosystem's adaptation to the disturbance as well as species population dynamics and ecosystem state (productivity, biomass), but also attributes describing the status of functional diversity that links the status of biodiversity to ecosystem functions. These data can be obtained from the flexible-trait DGVMs, high-resolution remote sensing, but requires extensive field-data for capturing changes in species and population dynamics as well as functional or chemical traits. With the identified tipping behaviour of the EI metric(s),

additional data analyses of the processes and attributes listed in Table 2 allow the interpretation and explanation of multi-factorial characteristics that may underpin BD-TPs.

BD-TPs can be identified at a particular site or landscape, but also affect a larger region. To differentiate BD-TPs from disturbance effects, larger regions should show such tipping behaviour, meaning that migration or dispersal of the affected species cannot compensate local BD loss and thus recovery is delayed or fails. Starting from tipped EI metric(s) at the

species level, search algorithms can be applied to capture the spatial dimension of tipping to assess when and at which spatial dimension ecosystem-level EI metric are affected or have tipped as well. Here, landscape connectivity as an indicator of structural EI could be a pre-condition for tipping compositional EI metric. Such behaviour could be investigated using techniques from percolation theory. In a next step, algorithms would be applied to then detect cascading effects of BD-TPs due to single or several tipped EI metric(s) (Dekker et al., 2018). It is possible that tipped structural and compositional EI

also affect functional EI metric which result from changes in disturbance regimes (Table 2). Such cascading effects can occur over time, i.e. different EI metric combinations tip due to the initial BD-TP, or at a different location or region through spatial connection of both affected regions. Because it is difficult to define functions or attributes that might cause BD-TPs a priori, an open search algorithm has the advantage that it is flexible in identifying where and under which conditions BD-TPs might occur and what the temporal and spatial dimension of such tipping might be. Drawing causal diagrams from driver and

effect relationships could possibly allow to identify cascading effects by combining EI metrics following Rocha et al. (2018), but it remains open if the methods can be applied to the biome scale and to data-scarce regions. In that sense the identification of BD-TP is different from identifying tipping points in the Earth system where it is clear that the dimension of tipping has to affect large-scale elements in the Earth system, such as biomes, ocean circulation systems or ice sheets (cf. Lenton et al., 2008), and tipping those should feed back to climate.






### 2.3 Impacts of biodiversity tipping points on human well-being

ES are the direct and indirect contributions of ecosystems to human well-being, which are grouped into regulating, provisioning and cultural services (MEA, 2005; TEEB, 2018). Supply and demand of ES changes according to the specific social-ecological context of the study region. Because global ES demand is often not sustainable and causes thereby NV

decline or defaunation, trade-offs with other ES or ecosystem functions occur, which increases the risk for cascading regime shifts that affect ES (Rocha et al., 2018). For example, in the tropical and subtropical forests of South America, meat from cattle and agricultural crops, mainly used for feeding livestock, are the most important provisioning ES at the macro-economic scale (Balvanera et al. 2011), although these hardly contribute to improve the well-being of local communities, who may only derive indirect benefits from these activities (e.g. in terms of labour opportunities and income), while they

may be affected by cumulate impacts – or ES disservices (e.g. in terms of health related to chemical inputs in intensive soy bean cultivation, loss of land and biodiversity).

Applying such a global approach to the regional and local scale of the Dry Diagonal remains a challenge and is open to debate. Indeed the assessment of ES demand and provision (in terms of amount and type) therefore depend on which region, scale and actor group is considered. In this respect, the recent IPBES nature's contribution to people approach helps to make

more visible less tangible, culturally specific ES of importance for indigenous populations, traditional (subsistence) farmers, and more generally those that may not be so readily quantified (c.f. Diaz et al., 2018).

ES describe the coupling flow between the ecosystems and society within a social-ecological system (Erb, 2012; Hummel, 2008; Liehr et al., 2017). Changes in EI affect the coupling flow and thus the ES supply. Land-use intensification in the Argentinian Dry Chaco can promote social conflicts among landholders with different preferences for ES (Mastrangelo and

Laterra, 2015). Using participatory approaches can here help to identify and describe land tenure conflicts in the context of different cultures of social and environmental interests (Seghezzo et al., 2017). Here, it is therefore essential to establish the link between tipped EI metric(s) and affected ES supply to be able to describe the potential risk for trade-offs with ES demand of the different actors in the SES (Fig. 3, 4[th] column).

With the obtained knowledge on tipped EI metric(s) and the mechanisms (processes and attributes affected), how the supply

of a specific ES is affected can be described. To analyse this link requires combinations of site or region-specific information on ES demand and detailed case studies that unravel the complexity of the social-ecological contexts at hand and illustrate emblematic local communities, i.e. their social structure, lifestyle, world views, degree of integration in the dominant culture and market economy, type of production system, customary legislation and local governance, to name but a few important dimensions that will influence the vulnerability and resilience of local communities to biodiversity tipping over the long

term. The key methodological challenge here is to collect sufficient, complementary information on how ES demand and related social-ecological implications may change. This requires close communication between different research teams and involved stakeholders, as well as iterative test phases to refine which EI indicator combination are more likely to occur the ES demand they may relate to.

At the same time, wider social-ecological implications of BD-TP for ES can also imply distal linkages which can be critical

for the export-oriented economies of, e.g. Argentina and Brazil. ES assessments, however, often overlook distant, diffuse and delayed impacts on ES resulting from social-ecological teleconnections. These effects are sometimes called off-site effects, displacement effects or off-stage burdens (Seppelt et al. 2011, Pascual et al. 2017). Applied to our case, local BD-TP and related ES changes may lead to rapid and profound loss in ES in distant areas outside the Dry Diagonal. Potential distal linkages in ES need to be understood so that place-based policies that aim at solving local ES loss in the Dry Diagonal may

not result distant ES burdens that may be invisible to, and thus underestimated by, local stakeholder groups. Mapping potential distal linkages using qualitative information could help to avoid such situation and reduce the risk of wider social-



ecological tipping. To this end, Dialogue processes including local and regional experts, sectoral representatives (e.g. agriculture, nature conservation, water management, local and regional governance) and lay members of local communities play a key role in isolating and interpreting the implications of distal linkages in ES.


Changing human-nature relationships also imply new challenges for nature conservation and related policy-making. In order to formulate comprehensive policy and management approaches that can help avoiding BD-TP and consequently potential social tipping, an exhaustive analysis of past and current conservation policies and economic incentives in the different countries of the Dry Diagonal is required. This could comprise a critical revision of area-based conservation instruments,

ranging from top-down land use restriction and economic incentive schemes, to bottom-up, participatory conservation programmes and hybrid approaches (Lambin et al., 2014). Brazil has a long history of such policies including the Forest Code and legal reserves, but conservation requirements in the Dry Diagonal reach at best 35% of set-aside areas of private properties (Fehlenberg et al., 2017; Soares-Filho et al., 2014). Based on emblematic cases, an important outcome here would be to identify success and failure factors in legal enforcement and the implementation of management strategies. The

emergence of BD-TP can spatially differentially affect the Dry Diagonal, which will challenge the current conceptualisation of how protected areas should be designed (e.g. in terms of size requirements, ecosystem composition, landscape connectivity, protection status, land use zonation and restrictions). Understanding spatial patterns that may be associated with tipping in the Dry Diagonal can provide important insights to facilitate the designation protected areas that can accommodate future ecological dynamics and their enforcement.

The ability of some States to effectively enforce environmental legislation to prevent or control deforestation has been put under scrutiny in the Chaco region of Argentina, where provincial governments were apparently unable to adequately enforce the mandate of the national Forest Law and their own forest planning cartography (Volante and Seghezzo, 2018). The second option, 'economic incentives' based on encouraging sustainable behaviours by positive, typically financial incentives, e.g. payments for ES (Wunder, 2015), the Brazilian market for forest quotas (Soares-Filho et al., 2016), incentive

programs for sustainable production (Le Tourneau and Greissing, 2010) or ecological tax programmes which transfers funds to municipalities according to ecological indicators (Ring, 2008). The third option, 'supply chain agreements' (e.g. soy and beef moratoria established for the Brazilian Amazon, (Gibbs et al., 2015)), aim at incentivizing responsible practices and helping farmers to maintain access to global markets and secure the ability to compete at higher value levels. Integrated livestock-soybean rotation systems saves land and could to incentivized through the Cerrado Soy moratorium and avoid

further NV decline in the Cerrado (Nepstad et al., 2019). Such agreements once more underline the opportunity to spare land and maintain biodiversity and EI for regions such as the Dry Diagonal.

These policy instruments have been developed for the social-ecological context of the Amazon rainforest, e.g. Wunder (2015) and Gibbs et al. (2015), or constitute country-specific regulations such as Brazil's Forest Code (Soares-Filho et al., 2014) or Argentina's Forest zones (Volante and Seghezzo, 2018). They would have to be revisited and carefully discussed to

assess the applicability of the respective instrument to another country of the Dry Diagonal, while recognized a different social-ecological context if the instrument in question can be applied to avoid BD-TP. One option to capture multiple actors, instruments and social-ecological context is to develop so-called meta-models for the countries of the Dry Diagonal. They would allow the identification of the social-ecological and political context in each country and also allow the validation of the meta-model with policy-makers, scientists and key informants. These meta-model can be based on the efficiency

parameters of the different conservation instruments using data from existing literature, 2) be complemented by qualitative analysis of the economic, political and social barriers for the establishment of the conservation approaches and instruments, and 3) be developed as a baseline (e.g. business-as-usual scenario) or a scenario to explore different paths in policy change possible to avoid BD-TPs, while they can form the basis of policy recommendations.




## 3. Outlook

Implementing the methodological framework to identify and analyse changes in ecological integrity that could lead to biodiversity tipping points and impact ES still requires substantial amount of work. It starts with producing a consistent data set on NV decline that follows a coherent classification system for the Cerrado and Dry Chaco covering the time span since the onset of industrial agriculture in the region. It would then form the data basis for quantifying structural EI metrics to capture landscape fragmentation. Furthermore, biodiversity data need to be harmonized and standardized to quantify

compositional EI metrics. Here, the data basis differs very much among countries and organism groups for which a common protocol would have to be developed. To contextualize the socio-cultural and socio-political conditions leading to NV decline as well as understanding the implications of ES trade-offs requires close collaboration between complementary social-science approaches to better understand the link between BD-TP and ES trade-offs. To obtain quantitative and qualitative data on the societal implications of BD-TPs, participatory approaches that both capture expert and lay knowledge

in diverse formats are required, while thorough institutional and policy analysis are needed to provide a differentiated, contextualised understanding of the potential opportunities and barriers to social-ecological resilience. The suggested meta-models to identify the different actor groups and their interests, cultural preferences, knowledge and economic basis is one opportunity to include social and political science expertise in the co-design and implementation of the suggested methodological framework.

To answer scientific insight to guide the design of policy instruments and governance structures that may be effective in avoiding biodiversity tipping points in potentially affected areas should take into account (i) the assessed impact of BD-TP on the supply of regional ES, (ii) the insights on underlying mechanisms of ES trade-offs and distal linkages, and (iii) the valuation of the efficacy of conservation tools. Because each potential strategy and policy mechanism has limitations, different implementation scenarios need to be tested in order to identify an optimal policy mix for the Dry Diagonal. Further,

maps that indicate potential future risks of biodiversity and social-ecological tipping points in the Dry Diagonal, including the possible distal linkages, could help decision-makers visualize and prioritize land use zoning and conservation management programs. More precisely, to tackle the risk of biodiversity or social-ecological tipping and foster sustainable human-nature relationships, spatially differentiated, policy and conservation measures are needed at national or landscape scale that identify leverage points for intervention through locally appropriate, tailored policy instruments on a case by case

basis.

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





**Acknowledgements**

The authors acknowledge funding from the German Federal Ministry of Science and Education (BMBF) for the project "Managing risks of biodiversity tipping points in South America's deforestation frontiers - Biodiv4Future", grant number 01LC1718A. Cristina de la Vega-Leinert is grateful for the support from the German Research Foundation under the project No. (V659/2-1). We thank the organizers and participants of the workshops in Bogotá (Columbia), Tucúman (Argentina),

and Brasilia (Brazil) in 2017. Furthermore, we thank Fabiana Arevalos (Guyra Paraguay), Ralf Seppelt (UFZ Leipzig), Tobias Kümmerle (HU Berlin), Julieta Delcarre and Gregorio Gavier Pizarro from INTA Buenos Aires (Argentina) for their contribution to the methodological framework and concept of this paper.

**Figures**

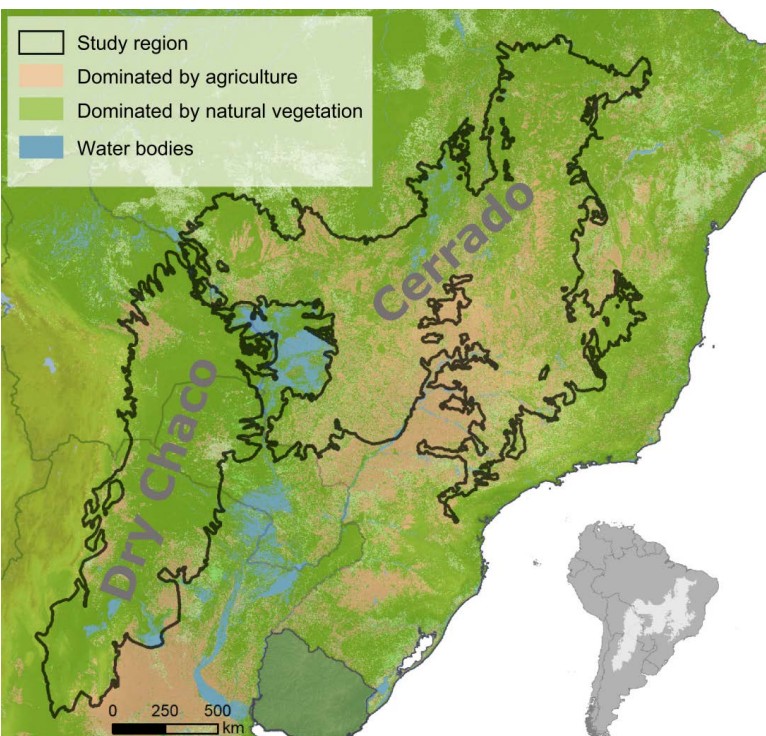

**Figure 1: Map of the Cerrado and Dry Chaco in the Dry Diagonal study region, showing the dominant land cover (areas dominated by agriculture in light red, areas dominated by natural vegetation in green, water bodies are shown in blue). Inserted map shows the location of the study region in South America. Land cover data are based on © ESA CCI Land Cover Data set (Defourny et al. 2014).**





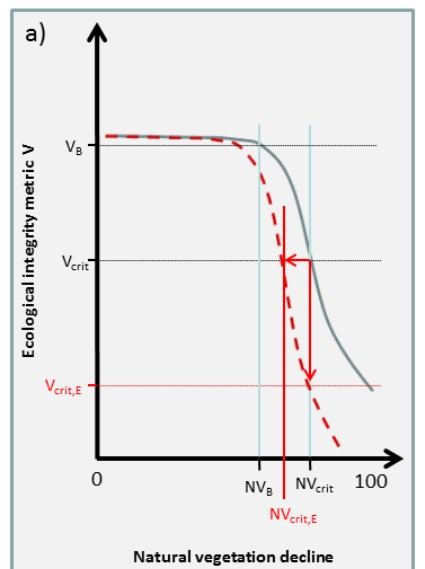 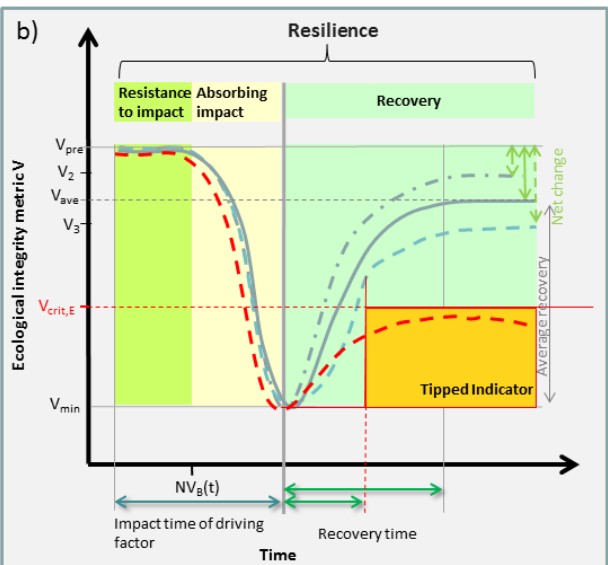


**Figure 2: Definition of ecological resilience and biodiversity tipping points: a) Systematic exploration and b) time-series analysis.**
**An ecosystem is exposed to NV decline forming a step function and disturbance impacts. NV decline is critical ($NV_{crit}$) when the**
**corresponding Ecological Integrity indicator V cannot recover from the impact ($V_{crit,E}$). An ecosystem is resistant when no break**
**point $NV_B$ at time t can be found. It is resilient to a specific impact when it is able to recover a substantial portion of the average EI**
**value ($V_{2,3} > V_{crit,E}$) after half of the average recovery time (b). Due to systems-internal feedbacks and dynamics recovery is often**
**incomplete and a net change remains, accounting for variability in ecosystem response and multiple ecosystem states. The**
**combination of NV decline and disturbance impacts can lead to internal feedbacks which further reduce the threshold for NV**
**decline ($NV_{crit,E}$ )**






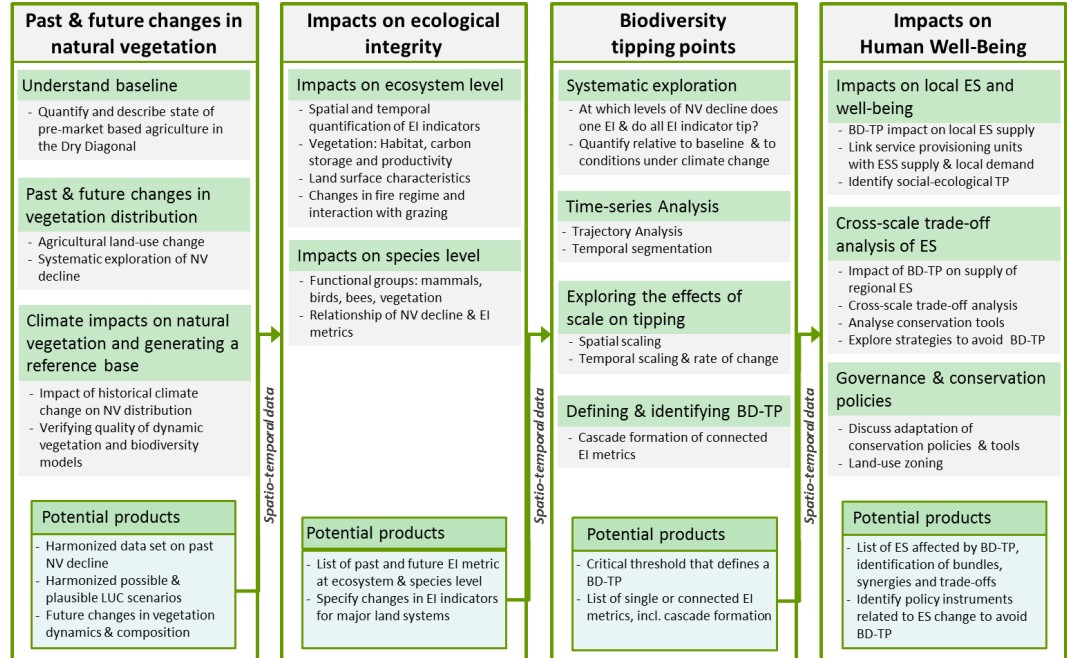

**Figure** 3**: Methodological framework to quantify past and future changes in natural vegetation (NV decline) based on agricultural expansion in the Dry Diagonal which then affects ecological integrity (EI) and increases the risk of crossing biodiversity tipping points (BD-TPs) and impacts ecosystem service provision (ES). Each column details the**

**required aspects to quantify NV decline (left column), impacts on EI components (second column), possible resulting biodiversity tipping points (third column) and impacts on human well-being (right column). Each column lists potential products which are necessary to assess the resulting consequences along the impact cascade. The potential products of the last column illustrate the wider use of the results in policy-making.**



**Tables**

**Table 1 Proposed indicators of ecological integrity, relating to its two hierarchical levels (species; and ecosystems) and its three components (structure; composition; and function).**

| | Structure | Composition | Function |
|---|---|---|---|
| Species | • Occurrence of key taxa (trees, birds, mammals, amphibians, butterflies)<br>• Viable populations of key taxa | • Richness of key taxa (trees, birds, mammals, amphibians, butterflies)<br>• Community turnover in key taxa (compared to natural systems) | • Occurrence of species with key functional traits (leaf and stem economics, predation, seed dispersal, etc.)<br>• Viable populations of species with key functional traits |
| Ecosystems | • Extent of natural vegetation (% woody vegetation)<br>• Occurrence and extent of key habitats (e.g. woodlands, grasslands, wetlands)<br>• Indicators of (landscape) fragmentation of natural habitats<br>• Structural connectivity | • Diversity of habitats (i.e., land covers)<br>• Diversity of phenological types / seasonality in vegetation<br>• Turnover in vegetation types (compared to natural systems) | • Primary productivity (e.g. GPP, NPP)<br>• Carbon stored in (above and belowground) biomass and soils<br>• Evapotranspiration<br>• Functional diversity and plant trait diversity<br>• Indicators of functional connectivity for key taxa (connectivity in suitable habitat, corridors) |





**Table 2: List of important processes or attributes that further detail phases of ecological resilience with natural vegetation decline (NV decline) as the primary, and water deficit, fire and grazing and the second order drivers. Colors correspond to phases of resilience shown in** Figure 2**b.**

| Driver | Processes/ Attributes important for resistance | Processes/ Attributes affected by impact | | Processes/Attributes decisive for recovery or contributing to tipping | | |
|---|---|---|---|---|---|---|
| | | Early | Late-impact | Short-term [Minutes to months] | Medium-term [Months to years] | Long-term [Years to decades] |
| NV decline | animal & plant meta-population size spp. range maintained with NV<Nv_crit Biomass | Meta-population size, spp. range affected with NV>NV_crit Biomass loss Reduced water storage | | reproduction rates migration of animals dispersal of plants | fragment size of natural vegetation (feeding & reproduction of animal population) Biomass Water storage | patch connectivity of natural vegetation Biomass Water storage |
| Water deficit | seed production seedling survival stomatal conductance growth rate phenological strategy | biomass allocation phenology | hydraulic architecture survival | stomatal regulation germination seed dormancy seed production carbon & water balance | Re-sprouting, biomass allocation trait shift growth rate | seed dispersal community composition richness & evenness functional diversity carbon storage stand density & |
| Fire | bark thickness canopy base height re-sprouting allocation to storage fuel production (biomass) micro-climate | cambium damage crown scorch live-fuel moisture | No. individuals scorched/ burned biomass consumed mortality rate | Re-sprouting from roots or crown germination post-fire mortality | see above plus post-fire mortality | see above |
| Herbivory | remaining biomass, old vs. new leave tissue protective traits (thorns, spines, chemical leaf traits) buds affected | buds affected seed number | buds affected carbon balance community change (palat- ability, annual vs. perrennial) transpiration & leaf area | regrowth from storage roots to replace lost tissue germination seed number | residual post- grazing mortality repeated grazing impact re-sprouting biomass allocation trait & community shift stand density & | seed production & dispersal community composition richness & evenness functional diversity carbon balance |

