# Peer review of "A social-ecological approach to identify and quantify biodiversity tipping points in South America's seasonal dry ecosystems"

_Biogeosciences, 2019_

## Referee Comment (RC1) · Anonymous Referee #1 · 7 Oct 2019

This manuscript "summarize[s] the outcome of [. . .] an interdisciplinary exercise in developing a methodological framework to identify and detect [biodiversity tipping points] for the deforestation frontiers of the Dry Diagonal." To develop such a framework is an ambitious and important task, and the case study a relevant one. However, I found that there are too many issues with this manuscript to recommend publication in Biogeosciences. To develop a framework as aimed for, both the thinking and the communication of it must be clear and unambiguous, otherwise the paper would just add confusion to the literature. I am sorry to write that I believe that this manuscript does not succeed in its task.

[Figure]

Here I list some examples of issues around the topic of resilience and tipping points that prevented me from understanding how the authors approach it. Please note that this list is non-exhaustive.

204-206 and 214 Ecological and engineering resilience have exact definitions. The fact that they are described here as "viewpoints" in "describing [a system's] behaviour" which "focus on" and "allow" things, and as "concepts" that can be "followed", suggests a lack of understanding or appreciation of this literature.

Indeed, in line 206: Bahn and Ingrisch (2018) – this is not an original reference for engineering and ecological resilience; please refer to Holling (1973). (This is later done in line 234, adding the problem of a lack of structure in introducing resilience. It seems as if different authors added different pieces of text without integrating them.)

218-220 Here the authors try to explain their definition of resilience, but not successfully: a change might lead to a modified ecosystem state, but it is not clear what kind of changes are meant. Also, how could a modified ecosystem state allow a system to have multiple stable states? It must be through feedbacks (see lines 207-208), but this is not clear in the text.

380 "One could also argue that the recovery of biodiversity after an impact is limited and remains below a critical threshold." I think it is unacceptable that a manuscript proposing a framework about biodiversity tipping points cannot provide clarity on the crucial matter of whether or not biodiversity recovery remains below a critical threshold. If it is up for argument, what is the value of the framework?

381 "Because single or several ecosystem or biodiversity components can contribute or cause tipping..." seems important, but is vague and unexplained.

386 What break-points?

409 "With the identified tipping behavior of the EI metric(s)..." Where has that been done?

412-413 So, apparently, the authors argue that biodiversity tipping points and disturbance effects are mutually exclusive. This is wrong.

416-417 I am clueless

455-460 I am even more clueless

In general, this manuscript is weakly written. The structure and flow of the text must be improved and the text appears hastily composed. There are grammatical errors, it is jargon- and acronym-laden, and often it is simply too vague (for me). Take, for example, this excerpt from lines 194-200: "The EI concept therefore allows us to adequately analyse the complex impact of NV decline on biodiversity while avoiding pitfalls of focusing solely on a single metric. By systematically exploring how respective EI metrics react to NV decline, the temporal and spatial scales of potential BD-TP can be identified. Because EI metrics can be linked to ES, the impact of BD-TP on (local and regional) livelihoods can be described. However, the societal implications of BD-TP and ES loss are often place and situation dependent. Therefore, to capture the consequences of BD-TP for the SES under study and derive generic insights of relevance for policy and management, detailed, contextualized, case study approaches, e.g. based on policy analysis and local ethnographies, play a critical role." Such writing is not clear, which is no exception in this manuscript. Therefore, I believe that this work is currently not valuable enough to readers outside the author team for publication.

The text contains sloppy mistakes. Below I list some examples from up to page 5 only. I stopped there because the text further on was even weaker, requiring more fundamental rewriting.

66 The first sentence mentions the Dry Diagonal without properly defining it. It is unclear whether it is limited to the Cerrado and Dry Chaco.

67 Dry Diagonal is singular, so "its biodiversity"

67-68 The statement that habitat losses increase the risk of biodiversity tipping points

is essential for this manuscript, yet no reference is provided

69 Systems do not have well-being

71 There is no reference for this sentence

94 Why "cf." – what's the contrast?

101 Note )(

105 I suppose you mean 2.5 times

110 Pardini is lacking a year

113 "it shares strong similarities in" – unclear what and what share similarities

116 in are

141 t should be at

165 The manuscript contains inconsistencies in the spelling of "south-eastern"

165 Kelvin or degrees Celsius, not degrees Kelvin

206 lists do not have dynamics

207 "van Nes et al. (37, p. 904)" – please fix. Also, I tried to look up this reference in the reference list, but it was absent. Apparently no check of consistency between the text and reference list has been done, so please do so

207 tipping points is plural, situation is singular

220 "the latter criterion" – referring to which list of criteria?

---

## Referee Comment (RC2) · Anonymous Referee #2 · 10 Oct 2019

This manuscript appears an attempt for a trans-disciplinary, integrated writing-up of a very courageous, extremely needed and useful project. Natural and social scientists, stakeholders, participatory workshop participants etc have all contributed to a very broad collection of ideas on how to tackle ecological problems in this ' Dry diagonal' of South America. Focus is on biodiversity loss, which is, no doubt, very substantial and worrying.

To write up results of such an exercise and to frame it in terms of tipping points in a multiple-stable states system that includes societal impacts and the role of societal actors, is laudable. The authors should be provided with every chance to make this a

success.

However, in my opinion, the present manuscript has failed in this ambition. Indeed I very much agree with the first sentence of the ' Outlook' section: " Implementing the methodological framework to identify and analyse changes in ecological integrity that could lead to biodiversity tipping points and impact ES still requires substantial amount of work."

Yes, a lot of work is still needed. This manuscript has at best been an inventory of the issues ranging from the definition and possible methodologies on tipping points, to many and very diverse examples of what is going on in the region. Its stated aim is to " summarise' the excercise to bring these together and to " review" the literature. To be honest, I do not think this manuscript can be regarded a summary: it is way to long and wide-ranging, too little focused, with too many and diverse examples. It is not a very good review either. Although many sources are cites, there is a lack of a systematic approach for such a review.

I also seem to recognise a multi-author excercise without proper leading editing. Several concepts are repeatedly addressed, in different ways with differing definitions. The very concept of tipping points and resilience is one example, the definition of ecosystem services is another.

On the tipping points, it is unclear whether a tipping point is defined as merely a (non-linear) decline in biodiversity, or whether is also needs to be poorly reversible. While figure 2 represents a pretty exact and technical definition of tipping points including irreversibility, this concept is hardly referred to elsewhere in the text. This definition is essential for both conservation policies and for detection methodologies.

A very useful concept, new to me, is the idea that the value of ecosystem services depends on who benefits from them, and that the impact of their decline can feed back on the system. This may be a very useful thread to build a more consistent article, and indeed a new methodology to approach tipping points and define pro-active policy

concepts. I would recommend rethinking and rewriting of this MS along these lines.

Detailed comments:

As i recommend complete rewriting, I don't think it is useful to comment in detail. However, a few issues:

- language needs thorough checking. There are many idiomatically poor and confusing sentences - PLEASE PLEASE PLEASE , do not use these abbreviations/acronyms. It does not help, it makes the reader stumble all the time. For example, this sentence: " Because EI metrics can be linked to ES, the impact of BD-TP on (local and regional) livelihoods can be described." is purely based on acronyms.. - in line 130, you mention a ' wide variety ' of communities, and then as an example only mention the quilombos. Surely, these are interesting communities but not the most important ones? - line 165: south-Eastern South America surely is not part of the dry diagonal (running NE-SW) is it? - avoid multiple, different and late definitions of the same thing in several places. - please try and keep sentences legible for everyone. I know writing style can differ a lot between natural and social scientists, but now many sentences are impenetrable. - line 436: I don't know everyting about ES, but do you really want to include intensive agriculture production in its definition? Would, for example, hydropower or mining products or land for housing/industry also be ES then? - line 510: where is 1)? - I think figure 3 and the tables are failrly useless. They seem a reflection of workshop flip-over notes rather than an attempt to concisely or comprehensibly summarise the issues.

-finally: the very last sentence leaves the reader with confusion: was this paper about exploring tipping points in biodiversity, or about policy measures?

Concluding: I am sorry to have to recommend that the MS in this form is unacceptable, for all the reasons mentioned. I would, however, very much welcome the authors to rethink and rewrite, maybe taking the concept of ES depending on cultural concept and feeding back to the system, as part of a multiple-stable states system. Try to use less examples, avoid each of you wanting to broadcast their own messages with

examples etc, take a rigid approach along a clearly defined methodology to develop a system and a policy framework.